# Large-Volume Focused-Ultrasound Mild Hyperthermia for Improving Blood-Brain Tumor Barrier Permeability Application

**DOI:** 10.3390/pharmaceutics14102012

**Published:** 2022-09-22

**Authors:** Hsin Chan, Hsin-Yun Chang, Win-Li Lin, Gin-Shin Chen

**Affiliations:** 1Institute of Biomedical Engineering, National Taiwan University, Taipei 100, Taiwan; 2Institute of Biomedical Engineering and Nanomedicine, National Health Research Institutes, Miaoli 35053, Taiwan; 3Institute of Biomedical Engineering, National Yang Ming Chiao Tung University, Hsinchu 300, Taiwan

**Keywords:** brain tumor, mild hyperthermia, transcranial focused ultrasound, phased array, multiple foci

## Abstract

Mild hyperthermia can locally enhance permeability of the blood-tumor barrier in brain tumors, improving delivery of antitumor nanodrugs. However, a clinical transcranial focused ultrasound (FUS) system does not provide this modality yet. The study aimed at the development of the transcranial FUS technique dedicated for large-volume mild hyperthermia in the brain. Acoustic pressure, multiple-foci, temperature and thermal dose induced by FUS were simulated in the brain through the skull. A 1-MHz, 114-element, spherical helmet transducer was fabricated to verify large-volume hyperthermia in the phantom. The simulated results showed that two foci were simultaneously formed at (2, 0, 0) and (−2, 0, 0) and at (0, 2, 0) and (0, −2, 0), using the phases of focusing pattern 1 and the phases of focusing pattern 2, respectively. Switching two focusing patterns at 5 Hz produced a hyperthermic zone with an ellipsoid of 7 mm × 6 mm × 11 mm in the brain and the temperature was 41–45 °C in the ellipsoid as the maximum intensity was 150 W/cm^2^ and sonication time was 3 min. The phased array driven by switching two mode phases generated a 41 °C-contour region of 10 ± 1 mm × 8 ± 2 mm × 13 ± 2 mm in the phantom after 3-min sonication. Therefore, we have demonstrated our developed FUS technique for large-volume mild hyperthermia.

## 1. Introduction

Brain tumors are classified as primary brain tumors and brain metastases by tumor cell origin. Although primary brain tumors are rare, they cause approximately 30% and 20% of cancer deaths, respectively, in children and young adults [1]. Furthermore, autopsy studies have suggested that the incidence of brain metastases is up to 40% in patients with cancer [2]. Clinical treatment methods include surgery, radiotherapy, chemotherapy and combination therapy [3]. However, the 5-year survival rate for patients with brain tumors is 35.8% in the United States [4], and patients with brain tumors suffer from the risk of complications, local recurrence after surgery [5,6] and possible side effects of radiation and chemo therapies [7].

After a brain tumor forms, the blood-brain barrier (BBB) within the tumor is impaired and termed a blood-tumor barrier (BTB). Permeability of the BTB to drugs was higher than that of the BBB but only 10% of lesions exhibited sufficient permeability in mice model [8]. Magnetic resonance imaging studies demonstrated similar findings that the contrast agent of gadolinium only penetrated to some of all experimental lesions [9,10]. These data are supported by the human study, which shows variable and ineffective uptake of preoperatively administered chemotherapeutic drugs into surgically resected tumors [11]. Consequently, the BTB remains sufficiently intact in most tumors to limit drug delivery and efficacy. Additionally, the BTB is heterogenous and increases interstitial fluid pressure in tumors, further hampering drug penetration [12]. Circulating chemo drugs for systemic therapy may cause cytotoxic side effects and even severe kidney and neurotoxicity [13,14]. Methods to enhance delivery of chemo drugs is needed for therapeutic efficacy and low-dose treatment of brain tumors.

Focused ultrasound (FUS) has been used to transiently disrupt the BBB based on nonthermal cavitation of microbubbles [15]. A few hemorrhage was observed in a rat neurooncological model after 1-MHz ultrasonic exposure with the acoustic pressure between 0.3–0.5 MPa [16] and none with the mechanical index smaller than 0.45 [17]. In addition, mild hyperthermia (40–45 °C) has also been shown to open the BBB for delivery of molecules and nanoparticles [18,19,20,21,22] without injection of the microbubbles. Mild hyperthermia could form intense cellular stress and result in transient BBB opening [23]. Morphological changes of individual endothelial cells by hyperthermia caused the loosened tight junctions and transported large molecules through the intercellular pathway [24]. After cessation of mild hyperthermia, loosened tight junctions commenced to recover, and the recovery rate depended on the heat dose and exposure time [25]. Thermal stress has also been shown to decrease the intratumoral pressure and increase nanoparticle accumulation [26,27]. Hyperthermia could modify elevated membrane fluidity and heat shock protein response of cancer cells [28], potentially improving drug delivery. The precisely molecular mechanism of hyperthermia BBB disruption is still unclear.

Hyperthermia techniques for BBB opening typically include radiofrequency [29,30], microwave [23,31], and laser [32,33]. However, these modalities have drawbacks of highly invasive operations, low spatial precision, inability to maintain the temperature of mild hyperthermia and/or superficial heating. FUS possesses advantages of noninvasive treatment, high spatial precision and deep penetration [34,35]. Recently, FUS has been used to perform mild hyperthermia in mouse brain tumors via the skull for applications in drug delivery [36,37,38]. FUS-induced temperature of 42–43 °C was maintained in brain tumor for approximately 10 min, and higher deposition of anticancer nanodrugs in the brain tumor and higher inhibition of the tumor growth were observed after 10-min hyperthermia. Glioblastoma is the most common primary brain cancer, and its diameter is usually within 5–10 cm at the time of diagnosis [39,40]. It will take many hours to perform hyperthermia of a typical glioblastoma if a single mm-scale hot spot is used to scan the whole cancer. Consequently, large volume hyperthermia is required. The methods of multiple simultaneous foci have been developed for volumetric ablation and hyperthermia in the body [41,42,43,44], but not applied to the brain with the skull barrier. In addition, it still lacks ultrasonic sonication strategies specific for volumetric hyperthermia of the brain although the clinical transcranial FUS system (ExAblate 4000 system, Insightec Ltd., Tirat Carmel, Israel) enables the formation of multiple simultaneous foci in the brain. Moreover, the clinical system consists of a 1024-element transducer and corresponding 1024-channel hardware, leading to complexity of operation and high cost. Therefore, the present study aimed to develop a FUS method for large-volume locally mild hyperthermia via the skull and reducing the channel count and system cost.

In the present study, we demonstrated the advancement of FUS transcranial hyperthermia technique at a temperature in the range of 41–45 °C, using the homemade helmet transducer with the human-use design. Numerical simulations of ultrasonic focusing, focal temperature and thermal dose were performed in the skull/brain model. A helmet phased array transducer and a skull/brain phantom were fabricated to verify the concept of the proposed FUS method.

## 2. Materials and Methods

### 2.1. Numerical Simulations

The skull model was reconstructed with CT images of a male head, which were acquired from Embodi3D’s biomedical image sources (www.embodi3d.com/about-us, accessed on 26 July 2019).

The outer surface and inner surface of the skull bone were defined as the water-skull interface and skull-brain interface, respectively (Figure 1a,b). The length, width and height of the skull model were 181.7 mm, 134.4 mm and 63.0 mm. The meshing resolution of the skull was one tenth of a wavelength, and the space between two skull interfaces were meshed by cubic interpolation with the increment of 0.625 mm. The frequency of the transducer was set to be 1.0 MHz. The spherical helmet transducer with the aperture diameter of 270 mm and the radius of curvature of 160 mm, similar to the commercial 1024-element phased-array transducer [45,46] was designed to ensure a sufficient power output without a high temperature rise in the transducer, low intensity impinging on the skull, and strong focusing on the target. The flat circular piezoelectric disc of 20 mm in diameters was employed because its electrical impedance was close to the magnitude of 50 Ω and the phase angle of 0° at 1.0 MHz, and hence no external impedance matching circuit was needed. The 114 elements were deployed in 5 rows to cover the spherical helmet (Figure 1c), where the gap between two rows was 25 mm and the pitch between two elements was 21.9 mm, 28.6 mm, 23.5 mm, 24.5 mm and 26.2 mm, respectively, in the bottom row, the 2nd row, the 3rd row, the 4th row and the top row. To inspect the influence of the element number on the performance of multiple simultaneous foci in numerical simulations, the other transducer model was constructed by 381 elements with the diameter of 10 mm deploying in 11 rows (Figure 1d), where the gap was 15 mm and all pitches were smaller than 15 mm. The curvature center of the transducer was set as the original point (0, 0, 0), and the bottom of the skull was on the xy plane at z = 0. Water, skull bone and brain tissue were assumed homogeneous mediums and their material properties are listed in the Table 1.

The wave reflection and refraction can occur at the water-skull interface and the skull-brain interface, leading to the shift and defocusing of the focus. We used the secondary source theory [49,50] and phase correction to minimize the above-mentioned problems. In the theory, the transducer is the primary source of vibration in the first medium and the point at the interface is assumed to be the secondary source for the next medium. Therefore, the points at the interface between any two mediums are regarded as the new source in the multilayer mediums. To begin with, the normal velocity *V_t_* of each point at the water-skull interface is calculated by using (1).
(1)$$V_{t}=\overset{N}{\underset{n=1}{\sum }}\frac{jk_{1}}{2\pi }u_{n}\frac{e^{-jk_{1}r_{n}}}{r_{n}}\left(1-j\frac{1}{k_{1}r_{n}}\right)T^{n}\cos\theta_{2}^{n}\Delta S_{n\;}$$
where the ultrasound transducer is divided into N point sources and the n represents the *n*th point source; k, u and ΔS are the wave number in water, the surface velocity of the point source at the transducer and the surface area of the nth discrete point source on the transducer, respectively; r is the distance between the point at the interface and the point source of the transducer; *T* is the transmission coefficient of the interface. *T* can be calculated by using (2).
(2)$$T=\frac{2\rho_{1}c_{1}\cos\theta_{1}}{\rho_{2}c_{2}\cos\theta_{1}+\rho_{1}c_{1}\cos\theta_{2}}$$
where $\rho_{i}$ and $c_{i}$ stand for the density and sound velocity of the medium *i* (*i* = 1, 2; 1: water, 2: skull bone); $\theta_{1}$ and $\theta_{2}$ are the incident and transmission angles of the wave at the water-skull interface and determined by the Snell’s law.

Furthermore, the normal velocity of each point at the skull-brain interface is obtained by using (1) and (2) when each point at the water-skull interface is set as the new source. Finally, the points at the skull-brain interface are chosen as the new source and the velocity u(x,y,z) of each point in the brain is calculated by using (3).
(3)$$u\left(x,y,z\right)=\frac{1}{2\pi }\overset{M}{\underset{m=1}{\sum }}V_{t}\frac{e^{j\left(\omega t-kr_{m}\right)}}{r_{m}}\Delta S_{m}$$
where the inner skull surface is divided into M point sources;$\;V_{t}$ is the normal velocity at the skull-brain interface; ω, k and $\Delta S_{m}$ are the angular frequency, the wave number in the brain and the surface area of the point source on the interface, respectively; r is the distance between the point in the brain and each point source at the skull-brain interface.

For phase compensation of each element, we simplified that the propagation path of the wave was a straight line from the element to the desired focus via mediums of water, skull and brain as shown in Figure 2. The A, B, and C denote the distance between the element and the outer surface of the skull, the distance between the skull surfaces, and the distance between the inner surface of the skull and the focus. The compensated phase of the element is equal to the remainder of the sum of the A_w_, B_w_, and C_w_ divided by 2π, where the A_w_, B_w_, and C_w_ are the values of A, B, and C multiplied by the wave number of water, skull, and brain, respectively.

To generate large-volume heating, multiple foci and switching focal patterns were designed in this study. When the peak temperature was up to 45 °C, ultrasonic sonication ceased for safety. The pseudo-inverse method [51] was used to estimate the phase difference between elements for constructive interference at the desired foci. The excitation source vector $\hat{u}$ of the element is a complex of $\left|\hat{u}\right|e^{i\left(\omega t+\varphi \right)}$, where φ is the phase and can be calculated by using (4).
(4)$$\hat{u}=H^{+}p$$
where $H^{+}$ is the pseudoinverse matrix of H; p is the complex acoustic pressure;
$$H=\left(\begin{array}{c} \begin{array}{c} \begin{array}{c} h_{11}\\ h_{21} \end{array}\\ \vdots \end{array}\\ h_{M^{\prime }1} \end{array}\begin{array}{c} \begin{array}{c} \begin{array}{c} h_{12}\\ h_{22} \end{array}\\ \vdots \end{array}\\ h_{M^{\prime }2} \end{array}\begin{array}{c} \begin{array}{c} \begin{array}{c} \ldots \\ \ldots \end{array}\\ \ddots \end{array}\\ \ldots \end{array}\begin{array}{c} \begin{array}{c} \begin{array}{c} h_{1N^{\prime }}\\ h_{2N^{\prime }} \end{array}\\ \vdots \end{array}\\ h_{M^{\prime }N^{\prime }} \end{array}\right)$$
$$h_{m^{\prime }n^{\prime }}=\frac{j\rho ck}{2\pi }\sum_{n^{\prime }=1}^{N^{\prime }}\frac{e^{-jk\left|r_{m^{\prime }}-r_{n^{\prime }}\right|}}{\left|r_{m^{\prime }}-r_{n^{\prime }}\right|}\Delta S_{n^{\prime }}$$

$M^{\prime }$ and $N^{\prime }$ represent the number of the designated focus (control points) and the number of phased-array elements, respectively; $r_{m^{\prime }}$ is the location of the control point and $r_{n^{\prime }}$ is the location of each grid on the single element; $\Delta S_{n\;}$ is the surface area of the n^th^ element. $M^{\prime }$ is typically smaller than $N^{\prime }$ and (4) is rewritten as (5).
(5)$$\hat{u}=H^{*}{\left(HH^{*}\right)}^{-1}p\;$$
where $H^{*}$ is the conjugate transpose of H. p can be obtained by Rayleigh-Sommerfeld diffraction integral Equation (6).
(6)$$p\left(x,y,z\right)=\overset{M}{\underset{m=1}{\sum }}\overset{N}{\underset{n=1}{\sum }}\frac{j\rho ck}{2\pi }\frac{e^{j\left(\omega t-kr_{mn}\right)}}{r_{mn}}u_{mn}\Delta S_{mn}$$

When multiple foci are formed, grating lobes occur simultaneously if the pitch of elements is larger than a half wavelength for the phased array transducer. Amplitude apodization of the phased array is performed to avoid the grating lobes in this study [52]. The additional phase ∅ weighted to the original phase of each element can be calculated by using (7).
(7)$$\emptyset =\left(M^{\prime }-1\right)\times \left(2\pi /M^{\prime }\right)$$

The temperature and thermal dose distributions are important indices to evaluate the efficacy of thermal therapy. The bio-heat transfer Equation (8) is typically used to estimate the temperature *T* distribution of the tissue [53,54].
(8)$$\rho_{t}c_{t}\frac{\partial T}{\partial t}=k\nabla^{2}T-w_{b}c_{b}\left(T-T_{ar}\right)+Q\;$$
where $\rho_{t}$ and $c_{t}$ are the density and specific heat of the tissue such as the brain, respectively. $k\nabla^{2}T$ means the thermal diffusion (k: thermal conductive coefficient of the tissue, $\nabla^{2}\;$ is discrete spatial Laplacian), $w_{b}$ is the blood perfusion rate, $c_{b}$ is the specific heat of blood, $T_{ar}$ is the temperature of arterial blood and Q is the energy absorbed by the tissue.
(9)$$Q=2\alpha I=\alpha \;\frac{{\left|p\left(x,y,z\right)\right|}^{2}}{\rho_{t}c}$$
where α, I and c are the attenuation coefficient, acoustic intensity on the tissue and sound velocity of the tissue, respectively.

The thermal dose (TD) of the treated tissue is defined as (10).
(10)$$\mathrm{TD}=\int_{t_{0}}^{t_{f}}R^{\left(T-43\right)}\;dt\;\{\begin{array}{c} R=2,\;\;for\;T\geq 43\;{}^{\circ}C\\ R=4,\;\;for\;37\;{}^{\circ}C<T<43\;{}^{\circ}C \end{array}$$
where $t_{0}\;$ and $t_{f}$ represent the start time and end time of ultrasonic sonication.

We used the software MATLAB R2017a in all simulations. For the simulations, the surface velocity of the transducer was an input variable, and its initial value was 1 mm/s. The operation frequency of the transducer was 1 MHz. We designed two focusing patterns and set the $M^{\prime }$ to 2 for each pattern. Two foci were created simultaneously when all elements of the phased array transducer were driven with their individual phase estimated by the pseudo-inverse method and phase correction. Two pairs of foci were alternated at the frequency of 5 Hz to perform the heating process. The initial temperature of the brain and blood was 37 °C. The calculated intensity was normalized in decibel (dB).

### 2.2. Fabrication of Phased-Array Transducer

The housing of the transducer was an acrylic spherical helmet with the thickness of 17.5 mm, the aperture diameter of 305 mm, and the radius of curvature of 160 mm. The helmet housing comprised a 76-mm window for support of the 114-element transducer and 114 identical 22-mm holes for placing elements (Figure 3a). The element was composed of the flat PZT 4-ceramic disc (Ceramic Transducer Design, Taiwan), and its diameter of 20 mm and thickness of 2 mm were determined to fit the impedance of 50 Ω and 0° at 1.0 MHz (Figure 3b). The positive electrode of the element was made on the same side with the negative electrode for convenient wire connection. A 10-m, 30-AWG, 50-Ω coaxial cable (D1370115BT, Wellshow Technology, Taiwan) was attached to the individual element. All cables were connected to the driving system through two 156-position DL connectors (ITT Cannon, Irvine, CA, USA). The elements were fixed and sealed in the hole of the helmet housing by epoxy resin (Dowsil 748, Dow Corning Corporation, Midland, MI, USA) to form the air-backed phased-array transducer (Figure 3c). All materials are non-ferroelectric for MRI compatibility. The electrical impedance of each element was measured by an impedance analyzer (Impedance Analyzers 6500B, Wayne Kerr Electronics, Bognor Regis, West Sussex, UK) to determine the resonant frequency and the corresponding impedance. Moreover, the output acoustic power of the element was measured by the ultrasound power meter (UPM-DT-1000PA, Ohmic Instruments, St. Charles, MI, USA), and its electroacoustic conversion efficiency was calculated by the acoustic power over the input electrical power. In all experiments, the transducer was driven by the power amplifier (Phased array generator 500-013, Advanced Surgical System, Tucson, AZ, USA).

### 2.3. Phantom Study

An acrylic skull model (Kyoto Kagaku) with the dimensions of 21 × 13 × 16.5 cm and the thickness of 3–5 mm and a 5 cm × 5 cm × 7 cm brick of hydrogel were adopted as the phantom of human skull and brain. The hydrogel was transparent and thermal-sensitive material and formulated with 150 mL of degassed water, 0.18 mL of Acrylicacid (Acros Organics, Geel, Belgium), 9 g of N-Isopropylacrylamide (Acros Organics, Geel, Belgium), 0.375 g of N,N’-Methylenebisacrylamide (Alfa Aesar, Haverhill, MA, USA), 0.195 g of Ammonium peroxodisulfate (Showa Chemical, Minato-ku, Tokyo, Japan) and 0.4 mL of N,N,N’,N’-tetramethylethylenediamine (Acros Organics, Geel, Belgium). White lesions could be formed at ultrasonic foci in the hydrogel when the localized temperature was over 41 °C. White lesions disappeared once the temperature was below 41 °C. The transducer and the skull model were deployed in accordance with their locations in the simulation. The center of the skull model was aligned with the center of the transducer in the z direction by the mechanical mechanisms. The bottom of the skull model and the curvature center of the transducer were arranged on the same plane (Figure 4a). The hydrogel was put beneath the skull model. All hyperthermia experiments were performed in a tank filled with degassed pure water with a temperature of 37 °C. The heating process was videotaped by the digital camera (NEX-3, Sony, Japan) to estimate the white area using the software Image J (1.50i, National Institute of Health, Bethesda, MD, USA). After sonication, the 0.2-mm needle thermocouple (Type T, HYPO, Omega Engineering, Stamford, CT, USA) was inserted into the center of the lesion to measure the temperature.

Numerical simulations of the ultrasonic focusing through the acrylic skull were performed to estimate the phase of each element in the phantom study. Images of the acrylic skull (Figure 4b) were acquired by the 3D scanning software (Geomagic Wrap, 3D Systems, San Francisco, CA, USA) for numerical modeling. Other conditions and the simulation procedure were the same as those in the previous description.

## 3. Results

The transducer did not naturally focus on the center of curvature (0, 0, 0) when the skull was in the propagation path (Figure 5a–c) in the simulation. With phase compensation of each element for skull aberration, waves were focused at (0, 0.05, 0) via the skull, and the focal zone of −6 dB intensity volume was such as an ellipsoid with the dimensions of 0.80 mm × 0.95 mm × 3.20 mm (Figure 5d–f). For the 381-element transducer, the similar results of simulations were obtained (not shown).

Two patterns of trans-skull two-foci focusing were simulated in the brain as shown in Figure 6. The designated focusing points were at (2, 0, 0) and (−2, 0, 0) for the focusing pattern 1, and at (0, 2, 0) and (0, −2, 0) for the focusing pattern 2. A summary of two focusing patterns for two transducers is listed in the Table 2.

The temperature and thermal dose of the brain was simulated as transcranial ultrasound with the peak intensity of 120 W/cm^2^ was sonicated for 3 min by the 114-element transducer. For the focusing pattern 1, the peak temperature increased from 37 °C to the maximum of 45 °C throughout ultrasound sonication (Figure 7a), and the size of the temperature contour of 41 °C was 8 mm × 4 mm × 10 mm at 3 min (Figure 7b–d). The maximum thermal dose was 6.57 CEM43 after ultrasound sonication (Figure 7e–g). For hyperthermia of the focusing pattern 2, the maximum temperature, the size of the 41 °C-contour volume and the maximum thermal dose were 44.8 °C, 5 mm × 7 mm × 8 mm and 5.48 CEM43 at the end of ultrasound sonication (not shown).

Switching the focusing pattern 1 and 2 at 5 Hz to heat the brain was simulated, and the temperature and thermal dose were obtained as shown in Figure 8. The trans-skull ultrasound with the peak intensity of 150 W/cm^2^ was sonicated for 3 min by the 114-element transducer. Peak temperature increased by 7.8 °C with in one minute and gradually to the maximum of 44.8 °C at 3 min (Figure 8a). The heating volume above 41 °C was 7 mm × 6 mm × 11 mm, and the maximum thermal dose was 5.48 CEM43 at 3 min (Figure 8b–g).

Impedance analyses showed that the resonant frequency of the element was 1.0 ± 0.1 MHz, and the amplitude and phase of the impedance were 50 ± 20 Ω and 10° ± 8° at 1.0 MHz, respectively. In addition, the measured electroacoustic conversion efficiency of the element was 83.1% ± 5.4% in the electrical-power range of 4.1 W to 29.5 W.

In the phantom experiments, ultrasound with the phase of the focusing pattern 1 and the total electrical power of 90 W produced two white lesions observed at 68 s (Figure 9b). Two lesions were horizontally symmetrical to the z axis, and the distance between them was measured to be 4 mm approximately. After the horizontal lesions disappeared, using the phase of the focusing pattern 2 and the electrical power of 90 W, two new lesions were formed at 65 s and vertically symmetrical to the z axis as shown in Figure 9d, where the oblique bird’s-eye view picture was taken to show two separate lesions. The distance between two lesions was about 4 mm. For large-volume mild hyperthermia, one of three experiments showed that the lesion of 4 mm × 3 mm × 5 mm was observed at 1 min as the sonication parameters were the phase of the focusing pattern 1 and the focusing pattern 2 at the switching rate of 5 Hz and the total electrical power of 90 W (Figure 10a). The lesion gradually increased to 4 mm × 6 mm × 8 mm at 2 min and 9 mm × 8 mm × 12 mm at 3 min (Figure 10b,c). The measured temperature was 45.4 °C at the center of the lesion.

## 4. Discussion

According to the results of numerical simulations, the 114-element transducer can perform similar focal patterns to the 381-element transducer under the same aperture diameter, radius of curvature and driving signals in the study. The primary difference between them is that the 381-element transducer generates the side lobes with smaller intensity (−12 dB to −18 dB peak intensity) for the focusing pattern 1 due to the smaller inter-element pitch, compared with the 114-element transducer (−6 dB to −12 dB peak intensity). Nonetheless, the side lobes did not visibly interfere in the temperature and thermal dose distribution in the brain after transcranial ultrasound exposure by the 114-element transducer as shown in Figure 7.

Mild hyperthermia requires an appropriate temperature and thermal dose in the target region for heating efficacy and safety. The previous evidence suggests that acoustic hyperthermia does not damage the brain when the maximum temperature and thermal dose of the treated brain tissue are lower than 48 °C and 10 CEM43 (cumulative equivalent minutes at 43 °C), respectively [36,55,56]. The present study demonstrates that transcranial ultrasound with appropriate parameters can heat the brain at the temperature of 41–45 °C in the most period of sonication time and the accumulated thermal dose of the simulation is below the threshold of 10 CEM43. It is noteworthy that the treated region of 41–45 °C reaches 1 cm^3^ approximately within 3 min ultrasonic exposure. Continuously steering a mm-scale hot spot facilitates high temperature at the target tissue and damages the tissue. Moreover, the steering technique takes much time to treat a cm-scale tissue and causes heat accumulation on the skull. Switching two focusing patterns can uniformly heat a volumetric tissue and prevent high temperature at the center of the target tissue. Furthermore, the switching sonication is relatively fast to finish volumetric hyperthermia and has a low risk of skull burn, compared with the steering technique.

Temporal switching of two focusing modes was employed for uniform hyperthermia in the tissue volume [57]. Simulations show that only the focusing pattern 1 or the focusing pattern 2 heating results in the irregular contour of 41 °C and the concentrated thermal dose at foci when switching two modes heating can produce the quasi-ellipsoid hot zone and relatively uniform distribution of thermal dose. The study experimentally verifies that the switching technique can lower power requirements, decrease treatment time, lower the peak temperature of the sonication and create a uniform thermal dose over the phantom volume.

The operating frequency has been optimized to be 0.6–0.7 MHz for transcranial high-intensity focused ultrasound ablation [58,59]. The ratio of the acoustic power at focus to the change in temperature of the skull surface is maximized at 0.6–0.7 MHz and the ratio decreases to 0.3–0.6 of the maximum at 1.0 MHz. The electrical power of 300–800 W is typically used to induce a hot spot at temperature above 55 °C to cause thermocoagulative necrosis of the target tissue [60]. In contrast with high energy and high temperature applications, the present study illustrated transcranial mild hyperthermia with continuous focused ultrasound at 1.0 MHz. Relatively low power of 90 W was used in the study and the induced peak temperature of 45 °C may have the low risk of thermal hazards to the skull. Commercial systems that use multi-element array transducer can benefit from the findings in this study.

The developed transducer is a wide-aperture spherically helmet such as the transducer of the Exablate Neuro system, which may infer few increases in temperature of the skull due to each element generates low power/intensity onto its projecting skull area, particularly under mild-hyperthermia conditions. Moreover, a cooling system of water circulation is constructed in the clinical-use FUS system to maintain the temperature of the scalp and avoid skin burn. In the present study, we performed simulations of the acoustic field and thermal dose in two numerical models through the same heating method and carried out phantom ablation to verify the feasibility of the proposed heating method. The secondary source theory used in simulation has the limitation as the following. The distance between the transducer and the water-skull interface should be much larger than wavelength. The thickness of the skull must be larger than the wavelength. The interface of the skull must be larger than the beam width so that the wave diffraction has no effect on the power depositions. Although the experimental results are similar to the simulations in the model of the acrylic skull model and hydrogel, the density and velocity of the human skull are very different from the acrylic skull model and some limitations of the established heating method may occur for human applications. The acoustic impedance of the human skull is higher than that of the acrylic skull model, leading to higher reflection so that the input power of each element may be increased to perform mild hyperthermia in the brain. Furthermore, the skull density ratio between cancellous and cortical bone is also a critical factor. A low ratio causes less ultrasonic transmission and focalization via the skull and typical values > 0.3 are considered suitable for transcranial FUS treatment [61].

There are several limitations and unsolved issues in the present study. To begin with, the hemisphere phased array transducer consisted of only 114 elements could fulfill transcranial focusing and mild hyperthermia, but the range of dynamic focusing on the z direction and focus steering in the x and y directions need further study. Next, the measurement of the temperature was carried out only at the one site and not in real time during sonication due to the use of the invasive thermal coupler. Consequently, the contour of the isotherm and thermal dose were not obtained in the phantom study. Considering safety, clinically practical use and visualization of the thermal information, noninvasive medical imaging technologies are suggested to monitor temperature in real time such as MR and ultrasound thermometry [62,63]. Furthermore, only one switching rate and one operating frequency were investigated in the study. Optimization of the temporal switching rate needs further study and the use of the lower operating frequency such as 0.6–0.7 MHz may increase the uniform heating volume. Finally, the homogeneous skull model was used in the study. The heterogeneous skull model with the acoustic parameters calculated from CT data [64,65] may be used for precisely personalized simulations and treatment.

## 5. Conclusions

The method of focused ultrasound mild hyperthermia via the acrylic skull have been developed in the study. Two orthogonally complementary patterns of ultrasonic foci are generated by the pseudo-inverse method and temporal switching of two focusing modes is used to produce the uniform distribution of thermal dose. With the hemisphere phased array transducer, a cm-scale volume can be uniformly heated at 41–45 °C in the tissue phantom for 3 min. The developed technology can be integrated with the MRI system for clinical use and may potentiate the increase in delivery of anti-cancer drugs to brain tumors.

## Figures and Tables

**Figure 1 pharmaceutics-14-02012-f001:**
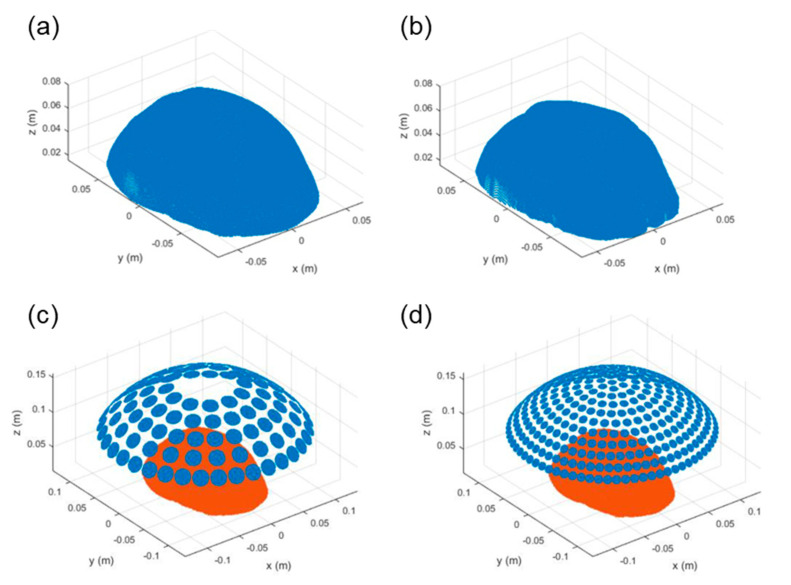
Numerical modeling of the human skull and the focused ultrasound transducers. The outer layer and inner layer of the skull were the interfaces between water and the skull bone (**a**) and between the bone and the brain (**b**), respectively. (**c**,**d**) show the models of 114-element and 381-element transducers above the skull.

**Figure 2 pharmaceutics-14-02012-f002:**
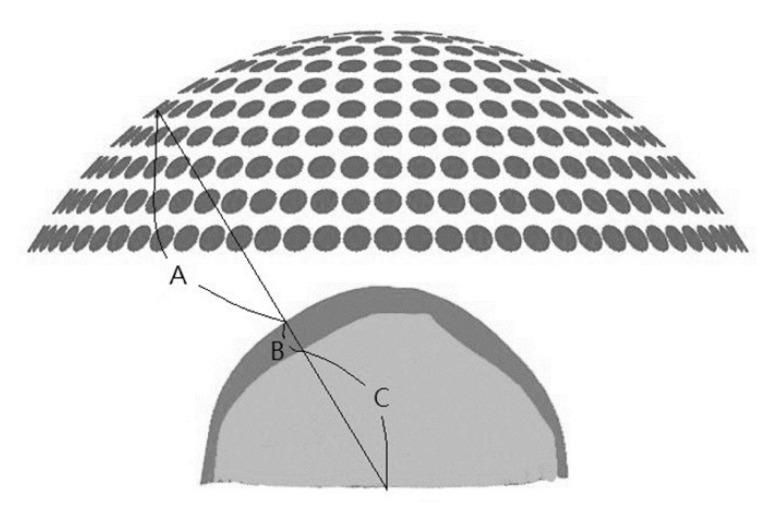
The hypothesized acoustic path of each element to the target brain tissue. The straight path was assumed to estimate the compensated phase of each element.

**Figure 3 pharmaceutics-14-02012-f003:**
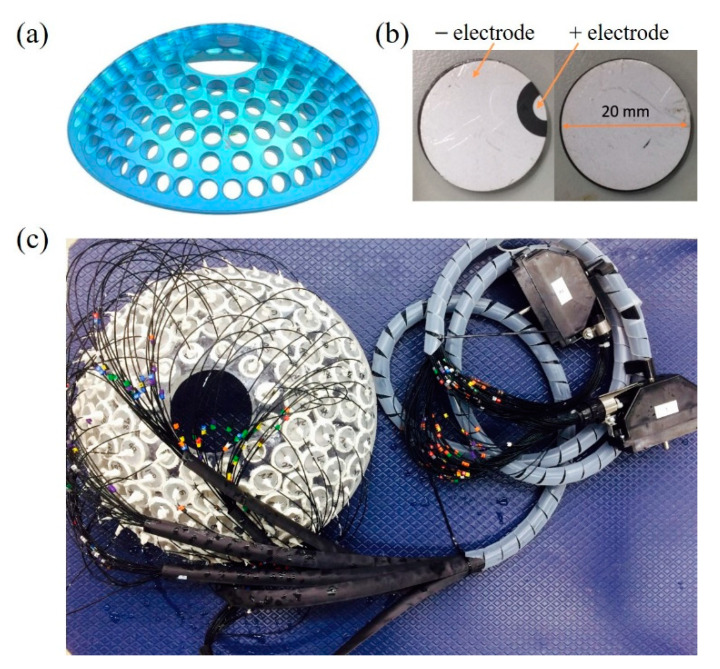
Configuration of the phased-array transducer. (**a**,**b**) show the schematic drawing of the helmet housing and a flat PZT-4 disc, respectively. The picture of the home-made phased-array prototype is shown in (**c**).

**Figure 4 pharmaceutics-14-02012-f004:**
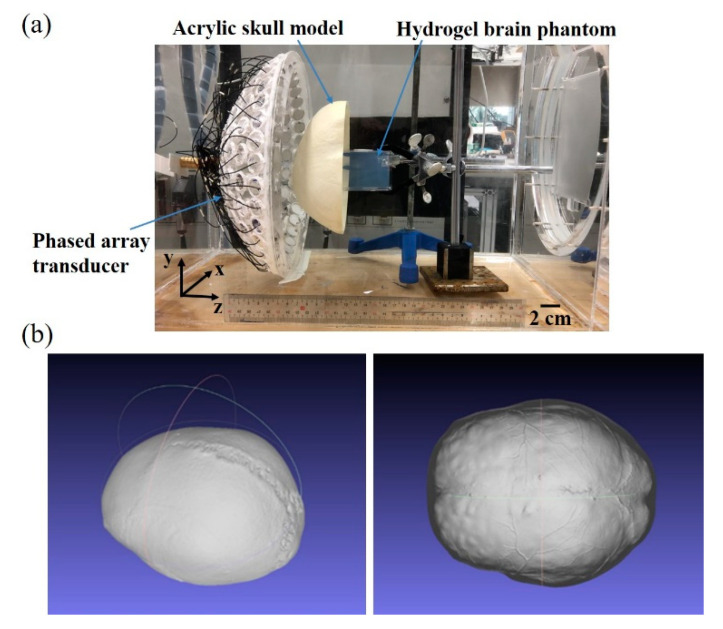
The experimental setup of the phantom study. The hemisphere transducer, the acrylic skull and the hydrogel phantom were aligned in the z direction (**a**). Images of the acrylic skull were taken for numerical modeling (**b**).

**Figure 5 pharmaceutics-14-02012-f005:**
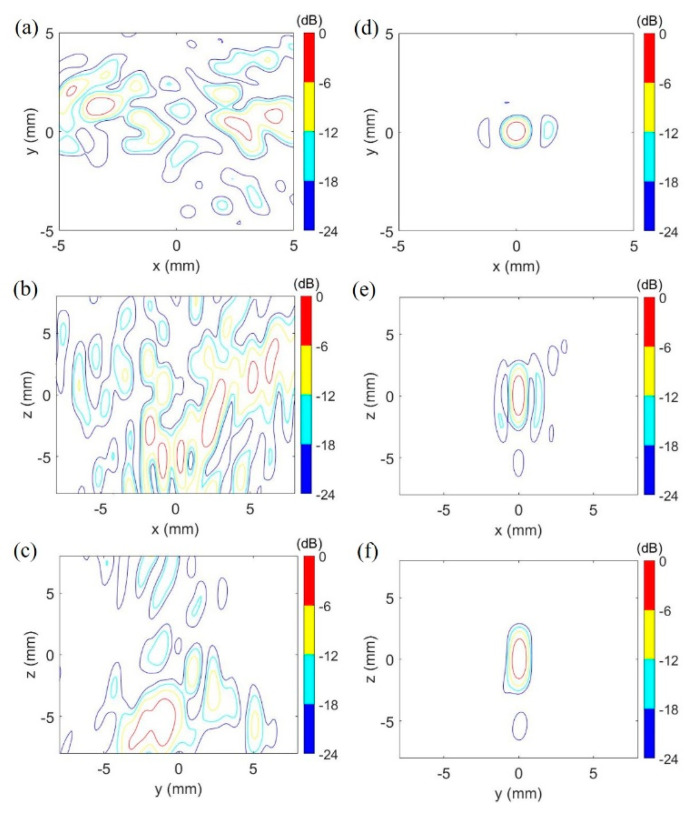
Simulated comparison between natural focusing and phase compensated focusing via the skull. The designated focusing point was the center of the curvature of the 114-element transducer at (0, 0, 0). The focus was distorted by the skull (**a**–**c**). After compensation of phases, the focal zone was a quasi-ellipsoid (**d**–**f**). The peak pressure was 8.60 MPa and 1.64 MPa, respectively, for the numerical model without the skull and with skull. dB: 20log (the intensity/peak intensity).

**Figure 6 pharmaceutics-14-02012-f006:**
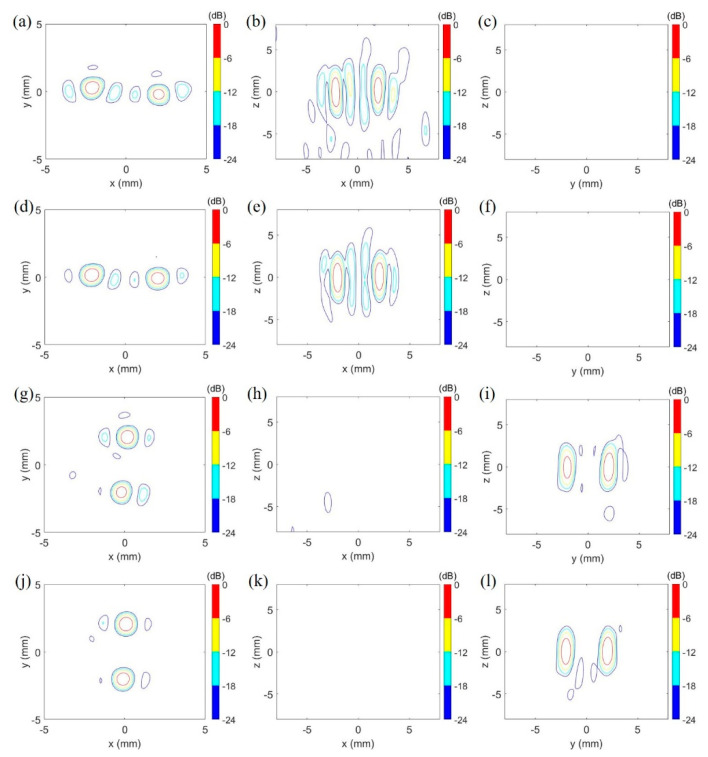
Focusing patterns of the 114-element and 381-element transducers with the focusing pattern 1 and the focusing pattern 2 in the simulation. (**a**–**f**) illustrate a pair of horizonal foci formed by the 114-element transducer and 381-element transducer with the focusing pattern 1, respectively. (**g**–**l**) shows the other pair of vertical foci produced by the 114-element transducer and 381-element transducer with the focusing pattern 2, respectively. The peak pressure was 9.70 MPa (no skull) & 1.95 MPa (via skull) and 9.70 MPa (no skull) & 2.18 MPa (via skull), respectively, for the focusing pattern 1 and 2 of the 114-element transducer. For the focusing pattern 1 and 2 of the 381-element transducer, the peak pressure was 8.15 MPa (no skull) & 1.63 MPa (via skull) and 8.15 MPa (no skull) & 1.82 MPa (via skull), respectively. dB: 20log (the intensity/peak intensity).

**Figure 7 pharmaceutics-14-02012-f007:**
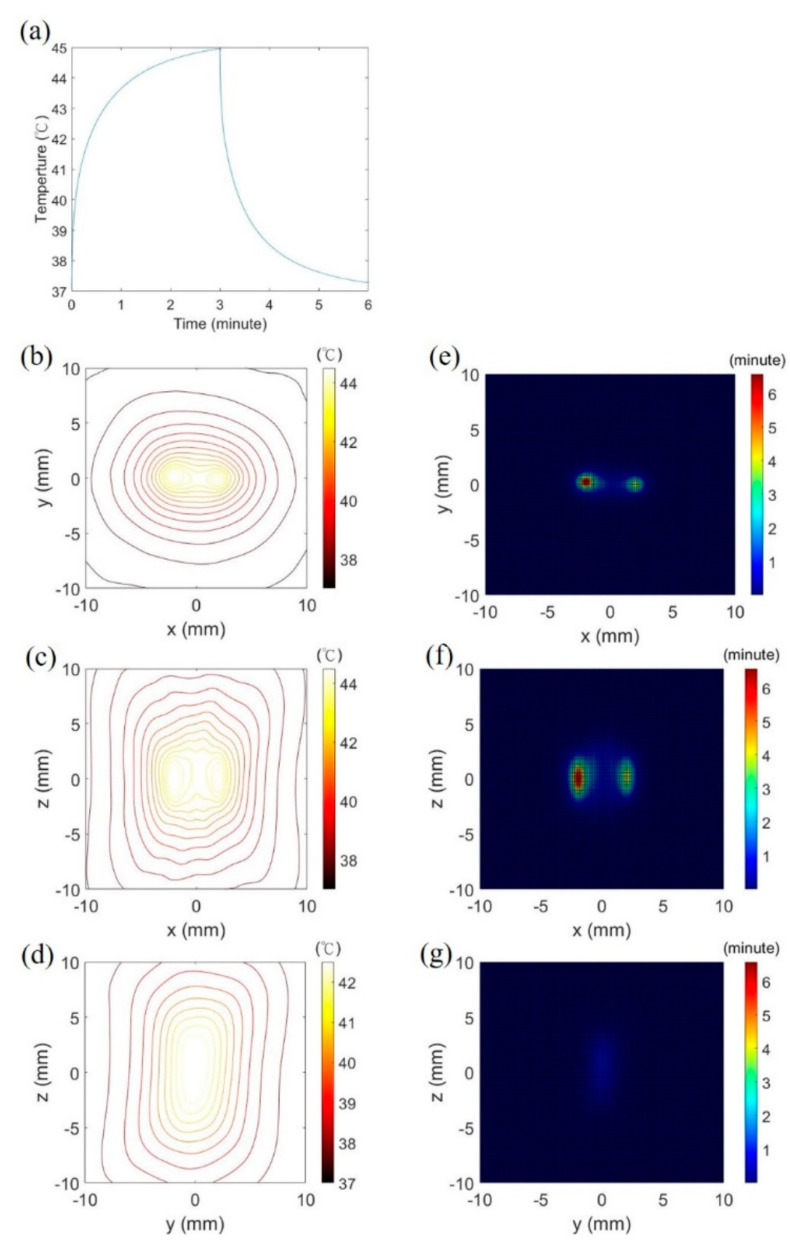
Thermal data of the 114-element transducer with the focusing pattern 1 in the simulation. The peak temperature increased from 37 °C to 41 °C at 11 s and 45 °C at 3 min (**a**). (**b**–**g**) delineate the spatial distribution of isotherm and thermal dose on three orthogonal plans, respectively.

**Figure 8 pharmaceutics-14-02012-f008:**
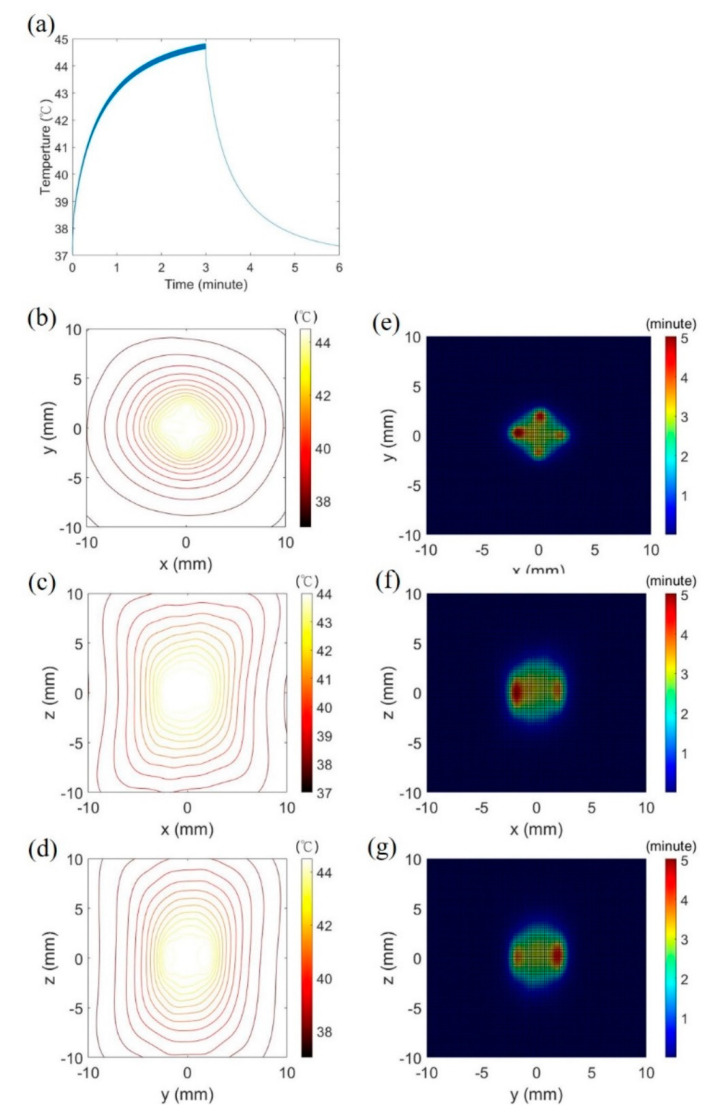
Thermal information obtained by switching the focusing pattern 1 and the focusing pattern 2 in the simulation. The peak temperature increased from 37 °C to 41 °C at 19 s and 44.8 °C at 3 min (**a**). (**b**–**g**) depict the spatial distribution of isotherm and thermal dose on three orthogonal plans, respectively.

**Figure 9 pharmaceutics-14-02012-f009:**
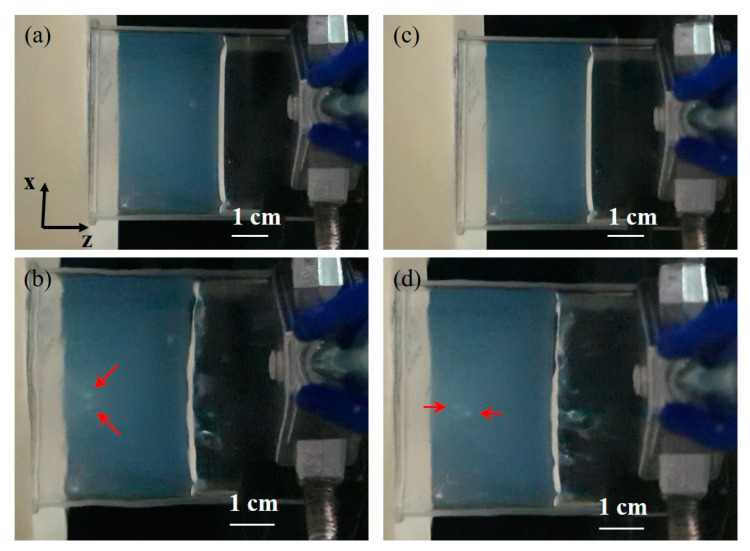
The phantom experiments using the focusing pattern 1 and focusing pattern 2 separately. (**a**,**b**) show the pictures of the phantom before and after sonication with the focusing pattern 1, respectively. Two white lesions were formed at two foci in the phantom and their location was similar to the simulations. For the focusing pattern 2, (**c**,**d**) show the phantom before sonication and two up and down white lesions produced by ultrasonic exposure, respectively. All photos were taken in the oblique bird’s-eye view for clear observation. The lesions were pointed out by red arrows.

**Figure 10 pharmaceutics-14-02012-f010:**
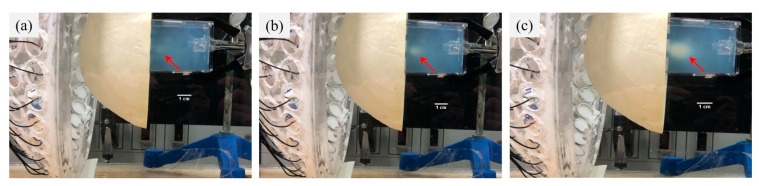
Ultrasonic hyperthermia in the phantom by switching the focusing pattern 1 and focusing pattern 2. The write lesion increased with the sonication time. (**a**–**c**) show the pictures of the phantom exposed to ultrasound at 1 min, 2 min and 3 min, respectively. The lesions were pointed out by red arrows.

**Table 1 pharmaceutics-14-02012-t001:** Parameters used in numerical simulations [47,48].

Parameters	Water	Skull Bone	Brain	Blood in Brain	Acrylic Skull	Hydrogel
ρ	1000	1796	1030	–	1040	1160
c	1500	2652	1545	–	1900	1505
α	0	176	4	–	185.4	5.8
c_t_ or c_b_	–	–	3640	3620	–	3365
*k*	–	–	0.528	–	–	0.60
*w_b_*	–	–	–	0.00833	–	–

ρ: density (kg/m^3^), c: velocity (m/s), α: attenuation at 1 MHz (np/m), c_t_ or c_b_: specific heat (J/kg·°C), *k*: thermal conductive coefficient (W/m·°C), *w_b_*: perfusion rate (kg/m^3^·s).

**Table 2 pharmaceutics-14-02012-t002:** Simulated results of two focusing patterns for two transducers.

	114-Element Transducer	381-Element Transducer
Focusing pattern 1	Focal zone of 0.75 mm × 0.70 mm × 3.00 mm at (2.05, −0.25, 0.25) and 0.85 mm × 0.90 mm × 3.45 mm at (−2.10, 0.25, 0.00) (Figure 6a–c)	Focal zone of 0.75 mm × 0.80 mm × 3.15 mm at (2.10, −0.10, 0.20) and 0.85 mm × 0.90 mm × 3.45 mm at (−2.10, 0.20, 0.00) (Figure 6d–f)
Focusing pattern 2	Focal zone of 0.85 mm × 1.00 mm × 3.30 mm at (0.20, 2.10, 0.00) and 0.65 mm × 0.80 mm × 2.55 mm at (−0.20, −2.05, 0.00) (Figure 6g–i)	Focal zone 0.80 mm × 1.05 mm × 3.20 mm at (0.10, 2.10, 0.10) and 0.80 mm × 0.09 mm × 3.05 mm at (−0.10, −2, 0.10) (Figure 6j–l)

## Data Availability

Not applicable.

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
