# Peer review of "Large-Volume Focused-Ultrasound Mild Hyperthermia for Improving Blood-Brain Tumor Barrier Permeability Application"

_pharmaceutics, 2022, doi:10.3390/pharmaceutics14102012_

Round 1
Reviewer 1 Report
This study presents design, development and investigation of ultrasound phased array transducer for transcranial hyperthermia delivery to the brain. Two phased array transducers were simulated while one array was fabricated and tested in phantom. While the development of helmet transducer arrays is an exciting field for brain applications such as opening blood brain barrier, rigorous tests need to be performed to characterize the devices and improve the efficiency of the transducers. This work is a step forward to evaluate such transducers for delivering mild hyperthermia to the brain, however, it lacks very important points which are mentioned below.
1) While the ExAblate 4000 system is in clinical use and can generate a sharp focus within the brain, why can’t the existing system be used for large volume hyperthermia if multiple simultaneous foci are produced with it. What is the rationale of developing another transducer and what are the clinical benefits?
22) The design of the transducer is not discussed in detail. Why were transducer discs as large as 20mm OD used? What were the dimensions of individual disc in the 381-element array? Why the authors chose 114 and 381 number of elements? What is the inter-element spacing between each disc? What is the steering range and how is it affected by the increase in the number of elements? How would be grating lobes avoided when all the elements are uniformly distributed in rows? What is meant by layers in the layout of the array, does it mean rows?
33) Why was the treatment time limited to 3 minutes? Usually, hyperthermia treatment is for longer durations, and it is important to test the devices at longer hyperthermia treatment times to rule out the effect on skull heating and skin burns.
44) The generation of multiple foci termed as ‘modes’ is incorrectly used. Usually ‘modes’ are used in concentric ring sector-vortex arrays, where mode # = N/2, N is the number of sectors. Thus, Mode1 will generate two foci, and Mode 2 will produce four simultaneous foci. This contrasts with the presented work, where Mode1 refers to two foci along one axis and mode2 refers to two simultaneous foci along another axis. This needs correction.
See T. Fjield and K. Hynynen, "The combined concentric-ring and sector-vortex phased array for MRI guided ultrasound surgery," in IEEE Transactions on Ultrasonics, Ferroelectrics, and Frequency Control, vol. 44, no. 5, pp. 1157-1167, Sept. 1997, doi: 10.1109/58.655641
55) Please discuss and cite references of similar work producing large volume hyperthermia/ablation with phased array transducers in other applications, some examples are below:
Hand, J. W., et al. "A random phased array device for delivery of high intensity focused ultrasound." Physics in Medicine & Biology 54.19 (2009): 5675.
Kim, Kisoo, et al. "Sonication strategies toward volumetric ultrasound hyperthermia treatment using the ExAblate body MRgFUS system." International Journal of Hyperthermia 38.1 (2021): 1590-1600.
Zubair M, Dickinson R. Calculating the Effect of Ribs on the Focus Quality of a Therapeutic Spherical Random Phased Array. Sensors (Basel). 2021 Feb 9;21(4):1211. doi: 10.3390/s21041211. PMID: 33572208; PMCID: PMC7915479
E. S. Ebbini and C. A. Cain, "A spherical-section ultrasound phased array applicator for deep localized hyperthermia," in IEEE Transactions on Biomedical Engineering, vol. 38, no. 7, pp. 634-643, July 1991, doi: 10.1109/10.83562
Some other comments are below:
P1 L19: Do you mean an ablation zone with an elliptical shape was produced? Correct the wording
P3 L177: Define ΔS
P5 L 199: Important ‘indices’ …
P5 L 211: Correct the denominator in eq. 9. Is it ρc ?
P6 L 227: Section 2.2 should be rewritten with explanation of the device design and images of the element distribution of both the transducers.
P6 L230: Support of which transducer? Did you mean ‘to accommodate an imaging transducer’?
P6 L233: What is meant by near 00, did you mean the phase was set to 00?
P6 L237: The elements were fixed and …
P6 L 238: Which epoxy was used to seal the transducers within the helmet?
P7 L258: This sentence is confusing, please rephrase.
P7 L 266: Explain how the geometric center of the transducer determined while the skull was in place. Was a hydrophone used to find the peak pressure at the geometric focus of the transducer?
P7 L269: Degassed water with a temperature of 37 0C
P7 L270: A digital camera
P7 L271: How long after the sonication the thermocouple was inserted to measure temperature? The phantom cools down fast and measurement should be taken within 15 secs after the power is shut down according to ESHO guidelines.
P8 Paragraph2: This paragraph needs correction. The modes are not previously defined and hence the reader does not know what I refers to. Second, the modes are incorrectly referred and instead another term such as ‘two simultaneous foci’ be used. You may refer to relevant papers such as below:
Zubair M, Dickinson R. Calculating the Effect of Ribs on the Focus Quality of a Therapeutic Spherical Random Phased Array. Sensors (Basel). 2021 Feb 9;21(4):1211. doi: 10.3390/s21041211. PMID: 33572208; PMCID: PMC7915479
Kim, Kisoo, et al. "Sonication strategies toward volumetric ultrasound hyperthermia treatment using the ExAblate body MRgFUS system." International Journal of Hyperthermia 38.1 (2021): 1590-1600.
P9 L330: Mention the time between switching modes as well
P9 L333: Rephrase to ‘Peak temperature increased by 7.80C with in one minute’
P13 L 391: It would be helpful to plot the sidelobe levels of both the arrays for a fair comparison
P13 L 397: … does not damage the brain
P14 L 443: What is the steering range of both the transducers?
P14 L463: Rephrase the sentence, it is confusing
Reviewer 2 Report
The authors present simulations and measurements of a new scheme for expanded volume transcranial hyperthermia using multiple foci. The manuscript is well organized, but there are a few concerns:
1. Considerably more detail is needed to help the reader understand the goal of the scheme (how large a volume is desired), how the results would be evaluated, and how the work is both novel and impactful to the field.
2. The description of the simulation approach needs more detail, including a discussion of assumptions and limitations of the numerical model employed.
3. The significance of the study seems overstated, given that the work was done on a single acrylic skull and in the absence of perfusing media. The authors have done good work, but need to moderate the conclusions about its impact until more work has been done.
4. Since the manuscript does not include any direct pharmaceutical or drug delivery results, there is an editorial assessment to be made whether the work is appropriate for Pharmaceutics, or it would be better placed in an ultrasonics journal.
Line specific comments follow:
L54. These references on mild hyperthermia mechanisms are between 13-31 years old. Can the authors confirm that there has been no relevant work in recent years?
L81. The authors should define their goal for 'relatively large' and explain why the expanded volume would be useful. This is a key part of explaining to the reader the novelty and impact of the work. The term large area should be replaced by large volume.
L96. More detail is needed here. Skulls support multiple wave types, and each can have sound speeds that spatially vary. If the model is only applied to acrylic, there are still multiple wave types present. The driving frequency should be specified here so the reader can make sense of the spatial mesh details that follow. Also - was convergence tested? Quarter wave sampling is extremely coarse and could incur large propagation errors. For example, k-wave (an existing validated code for transcranial simulations) recommends at least 1/10th wavelength sampling.
Table 1. Why did the brain-specific simulations not include perfusion?
L132. Clarify the phrase 'surface area of the point source'.
Eq 1. Clarify why T appears in Eq 1 (for water-skull interface) instead of a reflection term (T-1), since it is assumed that the point source propagates through water, to the skull and the total field at the interface is from the incident and skull-reflected fields. The primary assumption in this formulation seems to be that the skull is locally planar. Please clarify if this is the case and discuss the limitation of this assumption.
Eq 2. Follow this with Snell's law or what ever method was used to determine theta-2.
L165. Define the focal schemes being explored here, and if possible, illustrate them to help the reader understand what is being done. Also explain how the heating patterns will be judged - is there a goal for volume coverage? How is success determined?
L228. Give context for these specific choices. How do these compare to existing array geometries? If different, explain why. List the supplier for the PZT discs.
L271. Explain why (presumably to avoid thermocouple artefacts) the thermocouple was added after the sonication, and explain the likely measurement error that resulted. For lesion area estimation, why not section the phantom and directly measure the spot dimensions? Using a camera can result in aberration effects that distort size estimates unless done very carefully.
L283. Make clear whether this was a measurement or simulation result, both in the text and in figure captions.
L285. The focal shape recovery results need an additional column of images to show what the focal region looked like in the absence of the skull. How much amplitude was lost with the skull, and how much was recovered when phase corrections were applied . IN the figure caption, explain how the color scales were normalized.
L297. These results need to be placed in a table rather than written out in text.
Fig 6. Explain why c and f appear empty.
L362. Clarify if the locations were where they were intended to be, both laterally and axially.
L399. Be careful not to overstate the results. Only a single acrylic model was used. Measurements on actual skulls have not been described, so 'transcranial' should be replaced by 'acrylic transcranial model'.
L404. Were surface temperatures monitored? If not, what is the basis for the comment about reducing scalp/skull burning? More generally, please explain why it is advantageous to use a modal switching method rather than continuously steering through a similar volume? Both techniques would expand the heated volume.
L409. It is unclear why the rabbit thigh study is mentioned here. It would appear to belong in the introduction as evidence that modal switching can work for hyperthermia.
L435. Attenuation and impedance are two difference contributors to transmission loss- explain both in context of the proposed design.
L464. The comment about mechanical motion doesn't seem to follow directly from the presented work. If it is a suggestion than it probably belongs in the previous section.
L465. Phrases like 'potentiates increase in delivery' is not justified by the current work. Use 'may potentiate...' or something more conditional.
Reviewer 3 Report
Many researchers are concerned with the application of ultrasound hyperthermia for improving BTB permeability. The authors presented an interesting FUS technique dedicated for large-area mild hyperthermia in the brain. However, there are a few questions the authors should answer.
1. Figure 4 shows that the hydrogel brain phantom does not fill the entire acrylic skull model. The skull model is also filled with air. Does the signal from all transducers reach the phantom?
2. What is the ultrasonic wave attenuation of the phantom? The use of drugs (e.g. with nanoparticles) increases the absorption of ultrasound in tumor area. This can cause the temperature to rise faster and above 45 degrees. How to control this process so the temperature does not rise too much?
3. Phantom does not imitate the influence of blood perfusion on heating? Maybe some plastic pipes with fast-flowing water which mimic the blood vessels could put into the measurement setup, by which much more precise results could be obtained.
Reviewer 4 Report
The authors have represented a Research article entitled “Large-Area Focused-Ultrasound Mild Hyperthermia for Improving Blood-Tumor Barrier Permeability of Brain Tumors”. The manuscript was prepared well and it will be an important article for the researchers who intend to work in this field. There are some points that need to be discussed before acceptance.
- Please try to improve the Introduction section. In the Introduction, section authors need to explain in more detail similar types of reported work, with the novelty of the present work, limitations, and possibilities.
- Figure 1 needs to be modified.
- Please make a table with FDA-approved similar type devices their efficacy and limitations.
- Please change the title with more conclusion remarks with “application” terms
- What is the media between the device and target tissue to travel focused ultrasound? How the temperature and focus could be controllable? What is the penetration depth of ultrasound to reach the target tissue?
Round 2
Reviewer 1 Report
I am satisfied with the modifications.
Author Response
Thank you so much.
Reviewer 2 Report
The authors have improved their manuscript in response to questions from all reviewers. However, several of the original questions were not adequately addressed:
L139: The original question about mesh convergence agreement was not addressed. Would the results change if a finer grid was used? It is possible to get a 'good' answer with a fortunate choice of spatial sampling, but it may not work in a general case. It is common practice to test the convergence of numerical methods, and the authors really should address this point.
The original question about Eq 1 (now at L183) and the use of a transmission term at the water-skull interface rather than a reflection term was never answered, and this is a critical part of the formulation.
The original question about focal shape and magnitude recovery (regarding Figs 5 and 6) was never addressed in the revised manuscript. How would this system be used if transmitted pressures are not directly confirmed?
Also:
Figure 1 did not appear in the revision - only the caption was visible.
L93: In this last sentence of the pargraph, wasnt part of you goal to also reduce the channel count and system cost?
Reviewer 3 Report
The manuscript may be published.
Author Response
Thank you so much.